# Sample Efficient Path Integral Control under Uncertainty

**Yunpeng Pan, Evangelos A. Theodorou, and Michail Kontitsis**

Autonomous Control and Decision Systems Laboratory
Institute for Robotics and Intelligent Machines
School of Aerospace Engineering
Georgia Institute of Technology, Atlanta, GA 30332
{ypan37,evangelos.theodorou,kontitsis}@gatech.edu

## Abstract

We present a data-driven optimal control framework that is derived using the path integral (PI) control approach. We find iterative control laws analytically without a priori policy parameterization based on probabilistic representation of the learned dynamics model. The proposed algorithm operates in a forward-backward manner which differentiate it from other PI-related methods that perform forward sampling to find optimal controls. Our method uses significantly less samples to find analytic control laws compared to other approaches within the PI control family that rely on extensive sampling from given dynamics models or trials on physical systems in a model-free fashion. In addition, the learned controllers can be generalized to new tasks without re-sampling based on the compositionality theory for the linearly-solvable optimal control framework. We provide experimental results on three different tasks and comparisons with state-of-the-art model-based methods to demonstrate the efficiency and generalizability of the proposed framework.

## 1   Introduction

Stochastic optimal control (SOC) is a general and powerful framework with applications in many areas of science and engineering. However, despite the broad applicability, solving SOC problems remains challenging for systems in high-dimensional continuous state action spaces. Various function approximation approaches to optimal control are available [1, 2] but usually sensitive to model uncertainty. Over the last decade, SOC based on exponential transformation of the value function has demonstrated remarkable applicability in solving real world control and planning problems. In control theory the exponential transformation of the value function was introduced in [3, 4]. In the recent decade it has been explored in terms of path integral interpretations and theoretical generalizations [5, 6, 7, 8], discrete time formulations [9], and scalable RL/control algorithms [10, 11, 12, 13, 14]. The resulting stochastic optimal control frameworks are known as Path Integral (PI) control for continuous time, Kullback Leibler (KL) control for discrete time, or more generally Linearly Solvable Optimal Control [9, 15].

One of the most attractive characteristics of PI control is that optimal control problems can be solved with forward sampling of Stochastic Differential Equations (SDEs). While the process of sampling with SDEs is more scalable than numerically solving partial differential equations, it still suffers from the curse of dimensionality when performed in a naive fashion. One way to circumvent this problem is to parameterize policies [10, 11, 14] and then perform optimization with sampling. However, in this case one has to impose the structure of the policy a-priori, therefore restrict the possible optimal control solutions within the assumed parameterization. In addition, the optimized policy parameters can not be generalized to new tasks. In general, model-free PI policy search approaches

require a large number of samples from trials performed on real physical systems. The issue of sample inefficiency further restricts the applicability of PI control methods on physical systems with unknown or partially known dynamics.

Motivated by the aforementioned limitations, in this paper we introduce a sample efficient, model-based approach to PI control. Different from existing PI control approaches, our method combines the benefits of PI control theory [5, 6, 7] and probabilistic model-based reinforcement learning methodologies [16, 17]. The main characteristics of the our approach are summarized as follows

- It extends the PI control theory [5, 6, 7] to the case of uncertain systems. The structural constraint is enforced between the control cost and uncertainty of the learned dynamics, which can be viewed as a generalization of previous work [5, 6, 7].
- Different from parameterized PI controllers [10, 11, 14, 8], we find analytic control law without any policy parameterization.
- Rather than keeping a fixed control cost weight [5, 6, 7, 10, 18], or ignoring the constraint between control authority and noise level [11], in this work the control cost weight is adapted based on the explicit uncertainty of the learned dynamics model.
- The algorithm operates in a different manner compared to existing PI-related methods that perform forward sampling [5, 6, 7, 10, 18, 11, 12, 14, 8]. More precisely our method perform successive deterministic approximate inference and backward computation of optimal control law.
- The proposed model-based approach is significantly more sample efficient than sampling-based PI control [5, 6, 7, 18]. In RL setting our method is comparable to the state-of-the-art RL methods [17, 19] in terms of sample and computational efficiency.
- Thanks to the linearity of the backward Chapman-Kolmogorov PDE, the learned controllers can be generalized to new tasks without re-sampling by constructing composite controllers. In contrast, most policy search and trajectory optimization methods [10, 11, 14, 17, 19, 20, 21, 22] find policy parameters that can not be generalized.

## 2 Iterative Path Integral Control for a Class of Uncertain Systems

### 2.1 Problem formulation

We consider a nonlinear stochastic system described by the following differential equation

$$d\mathbf{x} = \big(\mathbf{f}(\mathbf{x}) + \mathbf{G}(\mathbf{x})\mathbf{u}\big)dt + \mathbf{B}d\boldsymbol{\omega}, \tag{1}$$

with state $\mathbf{x} \in \mathbb{R}^n$, control $\mathbf{u} \in \mathbb{R}^m$, and standard Brownian motion noise $\boldsymbol{\omega} \in \mathbb{R}^p$ with variance $\boldsymbol{\Sigma}_\omega$. $\mathbf{f}(\mathbf{x})$ is the unknown drift term (passive dynamics), $\mathbf{G}(\mathbf{x}) \in \mathbb{R}^{n \times m}$ is the control matrix and $\mathbf{B} \in \mathbb{R}^{n \times p}$ is the diffusion matrix. Given some previous control $\mathbf{u}^{old}$, we seek the optimal control correction term $\delta\mathbf{u}$ such that the total control $\mathbf{u} = \mathbf{u}^{old} + \delta\mathbf{u}$. The original system becomes

$$d\mathbf{x} = \big(\mathbf{f}(\mathbf{x}) + \mathbf{G}(\mathbf{x})(\mathbf{u}^{old} + \delta\mathbf{u})\big)dt + \mathbf{B}d\boldsymbol{\omega} = \underbrace{\big(\mathbf{f}(\mathbf{x}) + \mathbf{G}(\mathbf{x})\mathbf{u}^{old}\big)}_{\tilde{\mathbf{f}}(\mathbf{x}, \mathbf{u}^{old})}dt + \mathbf{G}(\mathbf{x})\delta\mathbf{u}dt + \mathbf{B}d\boldsymbol{\omega}.$$

In this work we assume the dynamics based on the previous control can be represented by Gaussian processes (GP) such that

$$\mathbf{f}_{\mathbb{GP}}(\mathbf{x}) = \tilde{\mathbf{f}}(\mathbf{x}, \mathbf{u}^{old})dt + \mathbf{B}d\boldsymbol{\omega}, \tag{2}$$

where $\mathbf{f}_{\mathbb{GP}}$ is the GP representation of the biased drift term $\tilde{\mathbf{f}}$ under the previous control. Now the original dynamical system (1) can be represented as follow

$$d\mathbf{x} = \mathbf{f}_{\mathbb{GP}} + \mathbf{G}\delta\mathbf{u}dt, \qquad \mathbf{f}_{\mathbb{GP}} \sim \mathcal{GP}(\boldsymbol{\mu}_f, \boldsymbol{\Sigma}_f), \tag{3}$$

where $\boldsymbol{\mu}_f, \boldsymbol{\Sigma}_f$ are predictive mean and covariance functions, respectively. For the GP model we use a prior of zero mean and covariance function $\mathbf{K}(\mathbf{x}_i, \mathbf{x}_j) = \sigma_s^2 \exp(-\frac{1}{2}(\mathbf{x}_i - \mathbf{x}_j)^{\mathrm{T}}\mathbf{W}(\mathbf{x}_i - \mathbf{x}_j)) + \delta_{ij}\sigma_\omega^2$, with $\sigma_s, \sigma_\omega, \mathbf{W}$ the hyper-parameters. $\delta_{ij}$ is the Kronecker symbol that is one iff $i = j$ and zero otherwise. Samples over $\mathbf{f}_{\mathbb{GP}}$ can be drawn using an vector of i.i.d. Gaussian variable $\Omega$

$$\tilde{\mathbf{f}}_{\mathbb{GP}} = \mu_f + \mathbf{L}_f\Omega \tag{4}$$

where $\mathbf{L}_f$ is obtained using Cholesky factorization such that $\mathbf{\Sigma}_f = \mathbf{L}_f \mathbf{L}_f^{\mathrm{T}}$. Note that generally $\Omega$ is an infinite dimensional vector and we can use the same sample to represent uncertainty during learning [23]. Without loss of generality we assume $\Omega$ to be the standard zero-mean Brownian motion. For the rest of the paper we use simplified notations with subscripts indicating the time step. The discrete-time representation of the system is $\mathbf{x}_{t+\mathrm{d}t} = \mathbf{x}_t + \boldsymbol{\mu}_{ft} + \mathbf{G}_t \delta \mathbf{u}_t \mathrm{d}t + \mathbf{L}_{ft}\Omega_t \sqrt{\mathrm{d}t}$, and the conditional probability of $\mathbf{x}_{t+\mathrm{d}t}$ given $\mathbf{x}_t$ and $\delta \mathbf{u}_t$ is a Gaussian $\mathrm{p}(\mathbf{x}_{t+\mathrm{d}t}|\mathbf{x}_t, \delta \mathbf{u}_t) = \mathcal{N}(\boldsymbol{\mu}_{t+\mathrm{d}t}, \mathbf{\Sigma}_{t+\mathrm{d}t})$, where $\boldsymbol{\mu}_{t+\mathrm{d}t} = \mathbf{x}_t + \boldsymbol{\mu}_{ft} + \mathbf{G}_t \delta \mathbf{u}_t$ and $\mathbf{\Sigma}_{t+\mathrm{d}t} = \mathbf{\Sigma}_{ft}$. In this paper we consider a finite-horizon stochastic optimal control problem

$$J(\mathbf{x}_0) = \mathbb{E}\Big[q(\mathbf{x}_T) + \int_{t=0}^{T} \mathcal{L}(\mathbf{x}_t, \delta \mathbf{u}_t)\mathrm{d}t\Big],$$

where the immediate cost is defined as $\mathcal{L}(\mathbf{x}_t, \mathbf{u}_t) = q(\mathbf{x}_t) + \frac{1}{2}\delta \mathbf{u}_t^{\mathrm{T}} \mathbf{R}_t \delta \mathbf{u}_t$, and $q(\mathbf{x}_t) = (\mathbf{x}_t - \mathbf{x}_t^d)^{\mathrm{T}} \mathbf{Q}(\mathbf{x}_t - \mathbf{x}_t^d)$ is a quadratic cost function where $\mathbf{x}_t^d$ is the desired state. $\mathbf{R}_t = \mathbf{R}(\mathbf{x}_t)$ is a state-dependent positive definite weight matrix. Next we show the linearized Hamilton-Jacobi-Bellman equation for this class of optimal control problems.

## 2.2 Linearized Hamilton-Jacobi-Bellman equation for uncertain dynamics

At each iteration the goal is to find the optimal control update $\delta \mathbf{u}_t$ that minimizes the value function

$$V(\mathbf{x}_t, t) = \min_{\delta \mathbf{u}_t} \mathbb{E}\Big[ \int_t^{t+\mathrm{d}t} \mathcal{L}(\mathbf{x}_t, \delta \mathbf{u}_t)\mathrm{d}t + V(\mathbf{x}_t + \mathrm{d}\mathbf{x}_t, t + \mathrm{d}t)\mathrm{d}t | \mathbf{x}_t \Big]. \tag{5}$$

(5) is the Bellman equation. By approximating the integral for a small $\mathrm{d}t$ and applying Itô's rule we obtain the Hamilton-Jacobi-Bellman (HJB) equation (detailed derivation is skipped):

$$-\partial_t V_t = \min_{\delta \mathbf{u}_t}(q_t + \frac{1}{2}\delta \mathbf{u}_t^{\mathrm{T}} \mathbf{R}_t \delta \mathbf{u}_t + (\boldsymbol{\mu}_{ft} + \mathbf{G}_t \delta \mathbf{u}_t)^{\mathrm{T}} \nabla_{\mathbf{x}} V_t + \frac{1}{2}\operatorname{Tr}(\mathbf{\Sigma}_{ft} \nabla_{\mathbf{xx}} V_t)).$$

To find the optimal control update, we take gradient of the above expression (inside the parentheses) with respect to $\delta \mathbf{u}_t$ and set to 0. This yields $\delta \mathbf{u}_t = -\mathbf{R}_t^{-1} \mathbf{G}_t^{\mathrm{T}} \nabla_{\mathbf{x}} V_t$. Inserting this expression into the HJB equation yields the following nonlinear and second order PDE

$$-\partial_t V_t = q_t + (\nabla_{\mathbf{x}} V_t)^{\mathrm{T}} \boldsymbol{\mu}_{ft} - \frac{1}{2}(\nabla_{\mathbf{x}} V_t)^{\mathrm{T}} \mathbf{G}_t \mathbf{R}^{-1} \mathbf{G}_t^{\mathrm{T}} \nabla_{\mathbf{x}} V_t + \frac{1}{2}\operatorname{Tr}(\mathbf{\Sigma}_{ft} \nabla_{\mathbf{xx}} V_t). \tag{6}$$

In order to solve the above PDE we use the exponential transformation of the value function $V_t = -\lambda \log \Psi_t$, where $\Psi_t = \Psi(\mathbf{x}_t)$ is called the *desirability* of $\mathbf{x}_t$. The corresponding partial derivatives can be found as $\partial_t V_t = -\frac{\lambda}{\Psi_t}\partial_t \Psi_t$, $\nabla_{\mathbf{x}} V_t = -\frac{\lambda}{\Psi_t}\nabla_{\mathbf{x}} \Psi_t$ and $\nabla_{\mathbf{xx}} V_t = \frac{\lambda}{\Psi_t^2}\nabla_{\mathbf{x}} \Psi_t \nabla_{\mathbf{x}} \Psi_t^{\mathrm{T}} - \frac{\lambda}{\Psi_t}\nabla_{\mathbf{xx}} \Psi_t$. Inserting these terms to (6) results in

$$\frac{\lambda}{\Psi_t}\partial_t \Psi_t = q_t - \frac{\lambda}{\Psi_t}(\nabla_{\mathbf{x}} \Psi_t)^{\mathrm{T}} \boldsymbol{\mu}_{ft} - \frac{\lambda^2}{2\Psi_t^2}(\nabla_{\mathbf{x}} \Psi_t)^{\mathrm{T}} \mathbf{G}_t \mathbf{R}_t^{-1} \mathbf{G}_t^{\mathrm{T}} \nabla_{\mathbf{x}} \Psi_t + \frac{\lambda}{2\Psi_t^2}\operatorname{Tr}((\nabla_{\mathbf{x}} \Psi_t)^{\mathrm{T}} \mathbf{\Sigma}_{ft} \nabla_{\mathbf{x}} \Psi_t) - \frac{\lambda}{2\Psi_t}\operatorname{Tr}(\nabla_{\mathbf{xx}} \Psi_t \mathbf{\Sigma}_{ft}).$$

The quadratic terms $\nabla_{\mathbf{x}} \Psi_t$ will cancel out under the assumption of $\lambda \mathbf{G}_t \mathbf{R}_t^{-1} \mathbf{G}_t^{\mathrm{T}} = \mathbf{\Sigma}_{ft}$. This constraint is different from existing works in path integral control [5, 6, 7, 10, 18, 8] where the constraint is enforced between the additive noise covariance and control authority, more precisely $\lambda \mathbf{G}_t \mathbf{R}_t^{-1} \mathbf{G}_t^{\mathrm{T}} = \mathbf{B}\mathbf{\Sigma}_{\omega}\mathbf{B}^{\mathrm{T}}$. The new constraint enables an adaptive update of control cost weight based on explicit uncertainty of the learned dynamics. In contrast, most existing works use a fixed control cost weight [5, 6, 7, 10, 18, 12, 14, 8]. This condition also leads to more exploration (more aggressive control) under high uncertainty and less exploration with more certain dynamics. Given the aforementioned assumption, the above PDE is simplified as

$$\partial_t \Psi_t = \frac{1}{\lambda}q_t \Psi_t - \boldsymbol{\mu}_{ft}^{\mathrm{T}} \nabla_{\mathbf{x}} \Psi_t - \frac{1}{2}\operatorname{Tr}(\nabla_{\mathbf{xx}} \Psi_t \mathbf{\Sigma}_{ft}), \tag{7}$$

subject to the terminal condition $\Psi_T = \exp(-\frac{1}{\lambda}q_T)$. The resulting Chapman-Kolmogorov PDE (7) is linear. In general, solving (7) analytically is intractable for nonlinear systems and cost functions. We apply the Feynman-Kac formula which gives a probabilistic representation of the solution of the linear PDE (7)

$$\Psi_t = \lim_{\mathrm{d}t \to 0} \int \mathrm{p}(\tau_t|\mathbf{x}_t) \exp\Big(-\frac{1}{\lambda}\big(\sum_{j=t}^{T-\mathrm{d}t} q_j \mathrm{d}t\big)\Big)\Psi_T \mathrm{d}\tau_t, \tag{8}$$

where $\tau_t$ is the state trajectory from time $t$ to $T$. The optimal control is obtained as

$$\mathbf{G}_t \delta \hat{\mathbf{u}}_t = -\mathbf{G}_t \mathbf{R}_t^{-1} \mathbf{G}_t^{\mathrm{T}} (\nabla_{\mathbf{x}} V_t) = \lambda \mathbf{G}_t \mathbf{R}_t^{-1} \mathbf{G}_t^{\mathrm{T}} \left( \frac{\nabla_{\mathbf{x}} \Psi_t}{\Psi_t} \right) = \mathbf{\Sigma}_{ft} \left( \frac{\nabla_{\mathbf{x}} \Psi_t}{\Psi_t} \right)$$

$$\Longrightarrow \hat{\mathbf{u}}_t = \mathbf{u}_t^{old} + \delta \hat{\mathbf{u}}_t = \mathbf{u}_t^{old} + \mathbf{G}_t^{-1} \mathbf{\Sigma}_{ft} \left( \frac{\nabla_{\mathbf{x}} \Psi_t}{\Psi_t} \right). \tag{9}$$

Rather than computing $\nabla_{\mathbf{x}} \Psi_t$ and $\Psi_t$, the optimal control $\hat{\mathbf{u}}_t$ can be approximated based on path costs of sampled trajectories. Next we briefly review some of the existing approaches.

## 2.3 Related works

According to the path integral control theory [5, 6, 7, 10, 18, 8], the stochastic optimal control problem becomes an approximation problem of a path integral (8). This problem can be solved by forward sampling of the uncontrolled ($\mathbf{u} = 0$) SDE (1). The optimal control $\hat{\mathbf{u}}_t$ is approximated based on path costs of sampled trajectories. Therefore the computation of optimal controls becomes a forward process. More precisely, when the control and noise act in the same subspace, the optimal control can be evaluated as the weighted average of the noise $\hat{\mathbf{u}}_t = \mathbb{E}_{\mathrm{p}(\tau_t | \mathbf{x}_t)} \left[ \mathrm{d} \boldsymbol{\omega}_t \right]$, where the probability of a trajectory is $\mathrm{p}(\tau_t | \mathbf{x}_t) = \frac{\exp(-\frac{1}{\lambda} S(\tau_t | \mathbf{x}_t))}{\int \exp(-\frac{1}{\lambda} S(\tau_t | \mathbf{x}_t)) \mathrm{d}\tau}$, and $S(\tau_t | \mathbf{x}_t)$ is defined as the path cost computed by performing forward sampling. However, these approaches require a large amount of samples from a given dynamics model, or extensive trials on physical systems when applied in model-free reinforcement learning settings. In order to improve sample efficiency, a nonparametric approach was developed by representing the desirability $\Psi_t$ in terms of linear operators in a reproducing kernel Hilbert space (RKHS) [12]. As a model-free approach, it allows sample re-use but relies on numerical methods to estimate the gradient of desirability, i.e., $\nabla_{\mathbf{x}} \Psi_t$, which can be computationally expensive. On the other hand, computing the analytic expressions of the path integral embedding is intractable and requires exact knowledge of the system dynamics. Furthermore, the control approximation is based on samples from the uncontrolled dynamics, which is usually not sufficient for highly nonlinear or underactuated systems.

Another class of PI-related method is based on policy parameterization. Notable approaches include PI$^2$ [10], PI$^2$-CMA [11], PI-REPS[14] and recently developed state-dependent PI[8]. The limitations of these methods are: 1) They do not take into account model uncertainty in the passive dynamics $\mathbf{f}(\mathbf{x})$. 2) The imposed policy parameterizations restrict optimal control solutions. 3) The optimized policy parameters can not be generalized to new tasks. A brief comparison of some of these methods can be found in Table 1. Motivated by the challenge of combining sample efficiency and generalizability, next we introduce a probabilistic model-based approach to compute the optimal control (9) analytically.

| | PI [5, 6, 7], iterative PI [18] | PI$^2$[10], PI$^2$-CMA [11] | PI-REPS[14] | State feedback PI[8] | Our method |
|---|---|---|---|---|---|
| Structural constraint | $\lambda \mathbf{G}_t \mathbf{R}_t^{-1} \mathbf{G}_t^{\mathrm{T}} = \mathbf{B} \mathbf{\Sigma}_\omega \mathbf{B}^{\mathrm{T}}$ | same as PI | same as PI | same as PI | $\lambda \mathbf{G} \mathbf{R}^{-1} \mathbf{G}^{\mathrm{T}} = \mathbf{\Sigma}_f$ |
| Dynamics model | model-based | model-free | model-based | model-based | GP model-based |
| Policy parameterization | No | Yes | Yes | Yes | No |

Table 1: Comparison with some notable and recent path integral-related approaches.

# 3 Proposed Approach

## 3.1 Analytic path integral control: a forward-backward scheme

In order to derive the proposed framework, firstly we learn the function $\mathbf{f}_{\mathbb{GP}}(\mathbf{x}_t) = \tilde{\mathbf{f}}(\mathbf{x}, \mathbf{u}^{old})\mathrm{d}t + \mathbf{B}\mathrm{d}\boldsymbol{\omega}$ from sampled data. Learning the continuous mapping from state to state transition can be viewed as an inference with the goal of inferring the state transition $\mathrm{d}\tilde{\mathbf{x}}_t = \mathbf{f}_{\mathbb{GP}}(\mathbf{x}_t)$. The kernel function has been defined in Sec.2.1, which can be interpreted as a similarity measure of random variables. More specifically, if the training input $\mathbf{x}_i$ and $\mathbf{x}_j$ are close to each other in the kernel space, their outputs $\mathrm{d}\mathbf{x}_i$ and $\mathrm{d}\mathbf{x}_j$ are highly correlated. Given a sequence of states $\{\mathbf{x}_0, \ldots \mathbf{x}_T\}$, and the corresponding state transition $\{\mathrm{d}\tilde{\mathbf{x}}_0, \ldots, \mathrm{d}\tilde{\mathbf{x}}_T\}$, the posterior distribution can be obtained by conditioning the joint prior distribution on the observations. In this work we make the standard assumption of independent outputs (no correlation between each output dimension).

To propagate the GP-based dynamics over a trajectory of time horizon $T$ we employ the moment matching approach [24, 17] to compute the predictive distribution. Given an input distribution over the state $\mathcal{N}(\boldsymbol{\mu}_t, \boldsymbol{\Sigma}_t)$, the predictive distribution over the state at $t + \mathrm{d}t$ can be approximated as a Gaussian $\mathrm{p}(\mathbf{x}_{t+\mathrm{d}t}) \approx \mathcal{N}(\boldsymbol{\mu}_{t+\mathrm{d}t}, \boldsymbol{\Sigma}_{t+\mathrm{d}t})$ such that

$$\boldsymbol{\mu}_{t+\mathrm{d}t} = \boldsymbol{\mu}_t + \boldsymbol{\mu}_{ft}, \quad \boldsymbol{\Sigma}_{t+\mathrm{d}t} = \boldsymbol{\Sigma}_t + \boldsymbol{\Sigma}_{ft} + \mathbb{COV}[\mathbf{x}_t, \mathrm{d}\tilde{\mathbf{x}}_t] + \mathbb{COV}[\mathrm{d}\tilde{\mathbf{x}}_t, \mathbf{x}_t]. \tag{10}$$

The above formulation is used to approximate one-step transition probabilities over the trajectory. Details regarding the moment matching method can be found in [24, 17]. All mean and variance terms can be computed analytically. The hyper-parameters $\sigma_s, \sigma_\omega, \mathbf{W}$ are learned by maximizing the log-likelihood of the training outputs given the inputs [25]. Given the approximation of transition probability (10), we now introduce a Bayesian nonparametric formulation of path integral control based on probabilistic representation of the dynamics. Firstly we perform approximate inference (forward propagation) to obtain the Gaussian belief (predictive mean and covariance of the state) over the trajectory. Since the exponential transformation of the state cost $\exp(-\frac{1}{\lambda}q(\mathbf{x})\mathrm{d}t)$ is an unnormalized Gaussian $\mathcal{N}(\mathbf{x}^d, \frac{2\lambda}{\mathrm{d}t}\mathbf{Q}^{-1})$. We can evaluate the following integral analytically

$$\int \mathcal{N}(\boldsymbol{\mu}_j, \boldsymbol{\Sigma}_j) \exp\left(-\frac{1}{\lambda}q_j\mathrm{d}t\right)\mathrm{d}\mathbf{x}_j = \left|\mathbf{I} + \frac{\mathrm{d}t}{2\lambda}\boldsymbol{\Sigma}_j\mathbf{Q}\right|^{-\frac{1}{2}} \exp\left(-\frac{1}{2}(\boldsymbol{\mu}_j - \mathbf{x}_j^d)^{\mathrm{T}}\frac{\mathrm{d}t}{2\lambda}\mathbf{Q}(\mathbf{I} + \frac{\mathrm{d}t}{2\lambda}\lambda\boldsymbol{\Sigma}_j\mathbf{Q})^{-1}(\boldsymbol{\mu}_j - \mathbf{x}_j^d)\right), \tag{11}$$

for $j = t+\mathrm{d}t, ..., T$. Thus given a boundary condition $\Psi_T = \exp(-\frac{1}{\lambda}q_T)$ and predictive distribution at the final step $\mathcal{N}(\boldsymbol{\mu}_T, \boldsymbol{\Sigma}_T)$, we can evaluate the one-step backward desirability $\Psi_{T-\mathrm{d}t}$ analytically using the above expression (11). More generally we use the following recursive rule

$$\Psi_{j-\mathrm{d}t} = \Phi(\mathbf{x}_j, \Psi_j) = \int \mathcal{N}(\boldsymbol{\mu}_j, \boldsymbol{\Sigma}_j) \exp\left(-\frac{1}{\lambda}q_j\mathrm{d}t\right)\Psi_j\mathrm{d}\mathbf{x}_j, \tag{12}$$

for $j = t + \mathrm{d}t, ..., T - \mathrm{d}t$. Since we use deterministic approximate inference based on (10) instead of explicitly sampling from the corresponding SDE, we approximate the conditional distribution $\mathrm{p}(\mathbf{x}_j|\mathbf{x}_{j-\mathrm{d}t})$ by the Gaussian predictive distribution $\mathcal{N}(\boldsymbol{\mu}_j, \boldsymbol{\Sigma}_j)$. Therefore the path integral

$$\Psi_t = \int \mathrm{p}\left(\tau_t|\mathbf{x}_t\right)\exp\left(-\frac{1}{\lambda}\left(\sum_{j=t}^{T-\mathrm{d}t}q_j\mathrm{d}t\right)\right)\Psi_T\mathrm{d}\tau_t$$

$$\approx \int ... \int \mathcal{N}\left(\boldsymbol{\mu}_{T-\mathrm{d}t}, \boldsymbol{\Sigma}_{T-\mathrm{d}t}\right)\exp\left(-\frac{1}{\lambda}q_{T-\mathrm{d}t}\mathrm{d}t\right)\underbrace{\int \mathcal{N}\left(\boldsymbol{\mu}_T, \Sigma_T\right)\underbrace{\exp\left(-\frac{1}{\lambda}q_T\right)}_{\Psi_T}\mathrm{d}\mathbf{x}_T}_{\Psi_{T-\mathrm{d}t}}\mathrm{d}\mathbf{x}_{T-\mathrm{d}t}...\mathrm{d}\mathbf{x}_{t+\mathrm{d}t}$$

$$\underbrace{\phantom{\approx \int ... \int \mathcal{N}\left(\boldsymbol{\mu}_{T-\mathrm{d}t}, \boldsymbol{\Sigma}_{T-\mathrm{d}t}\right)}}_{\Psi_{T-2\mathrm{d}t}}$$

$$= \int \mathcal{N}\left(\boldsymbol{\mu}_{t+\mathrm{d}t}, \boldsymbol{\Sigma}_{t+\mathrm{d}t}\right)\exp\left(-\frac{1}{\lambda}q_{t+\mathrm{d}t}\mathrm{d}t\right)\Psi_{t+\mathrm{d}t}\mathrm{d}\mathbf{x}_{t+\mathrm{d}t} = \Phi(\mathbf{x}_{t+\mathrm{d}t}, \Psi_{t+\mathrm{d}t}). \tag{13}$$

We evaluate the desirability $\Psi_t$ backward in time by successive computation using the above recursive expression. The optimal control law $\hat{\mathbf{u}}_t$ (9) requires gradients of the desirability function with respect to the state, which can be computed backward in time as well. For simplicity we denote the function $\Phi(\mathbf{x}_j, \Psi_j)$ by $\Phi_j$. Thus we compute the gradient of the recursive expression (13)

$$\nabla_{\mathbf{x}}\Psi_{j-\mathrm{d}t} = \Psi_j\nabla_{\mathbf{x}}\Phi_j + \Phi_j\nabla_{\mathbf{x}}\Psi_j, \tag{14}$$

where $j = t + \mathrm{d}t, ..., T - \mathrm{d}t$. Given the expression in (11) we compute the gradient terms in (14) as

$$\nabla_{\mathbf{x}}\Phi_j = \frac{\mathrm{d}\Phi_j}{\mathrm{d}\mathrm{p}(\mathbf{x}_j)}\frac{\mathrm{d}\mathrm{p}(\mathbf{x}_j)}{\mathrm{d}\mathbf{x}_t} = \frac{\partial\Phi_j}{\partial\boldsymbol{\mu}_j}\frac{\mathrm{d}\boldsymbol{\mu}_j}{\mathrm{d}\mathbf{x}_t} + \frac{\partial\Phi_j}{\partial\boldsymbol{\Sigma}_j}\frac{\mathrm{d}\boldsymbol{\Sigma}_j}{\mathrm{d}\mathbf{x}_t}, \text{ where } \frac{\partial\Phi_j}{\partial\boldsymbol{\mu}_j} = \Phi_j(\boldsymbol{\mu}_j - \mathbf{x}_j^d)^T\frac{\mathrm{d}t}{2\lambda}\mathbf{Q}(\mathbf{I} + \frac{\mathrm{d}t}{2\lambda}\lambda\boldsymbol{\Sigma}_j\mathbf{Q})^{-1},$$

$$\frac{\partial\Phi_j}{\partial\boldsymbol{\Sigma}_j} = \frac{\Phi_j}{2}\left(\frac{\mathrm{d}t}{2\lambda}\mathbf{Q}(\mathbf{I} + \frac{\mathrm{d}t}{2\lambda}\lambda\boldsymbol{\Sigma}_j\mathbf{Q})^{-1}(\boldsymbol{\mu}_j - \mathbf{x}_j^d)(\boldsymbol{\mu}_j - \mathbf{x}_j^d)^T - \mathbf{I}\right)\frac{\mathrm{d}t}{2\lambda}\mathbf{Q}(\mathbf{I} + \frac{\mathrm{d}t}{2\lambda}\lambda\boldsymbol{\Sigma}_j\mathbf{Q})^{-1}, \text{ and}$$

$$\frac{\mathrm{d}\{\boldsymbol{\mu}_j, \boldsymbol{\Sigma}_j\}}{\mathrm{d}\mathbf{x}_t} = \left\{\frac{\partial\boldsymbol{\mu}_j}{\partial\boldsymbol{\mu}_{j-\mathrm{d}t}}\frac{\mathrm{d}\boldsymbol{\mu}_{j-\mathrm{d}t}}{\mathrm{d}\mathbf{x}_t} + \frac{\partial\boldsymbol{\mu}_j}{\partial\boldsymbol{\Sigma}_{j-\mathrm{d}t}}\frac{\mathrm{d}\boldsymbol{\Sigma}_{j-\mathrm{d}t}}{\mathrm{d}\mathbf{x}_t}, \frac{\partial\boldsymbol{\Sigma}_j}{\partial\boldsymbol{\mu}_{j-\mathrm{d}t}}\frac{\mathrm{d}\boldsymbol{\mu}_{j-\mathrm{d}t}}{\mathrm{d}\mathbf{x}_t} + \frac{\partial\boldsymbol{\Sigma}_j}{\partial\boldsymbol{\Sigma}_{j-\mathrm{d}t}}\frac{\mathrm{d}\boldsymbol{\Sigma}_{j-\mathrm{d}t}}{\mathrm{d}\mathbf{x}_t}\right\}.$$

The term $\nabla_{\mathbf{x}}\Psi_{T-\mathrm{d}t}$ is compute similarly. The partial derivatives $\frac{\partial\boldsymbol{\mu}_j}{\partial\boldsymbol{\mu}_{j-\mathrm{d}t}}, \frac{\partial\boldsymbol{\mu}_j}{\partial\boldsymbol{\Sigma}_{j-\mathrm{d}t}}, \frac{\partial\boldsymbol{\Sigma}_j}{\partial\boldsymbol{\mu}_{j-\mathrm{d}t}}, \frac{\partial\boldsymbol{\Sigma}_j}{\partial\boldsymbol{\Sigma}_{j-\mathrm{d}t}}$ can be computed analytically as in [17]. We compute all gradients using this scheme without any numerical method (finite differences, etc.). Given $\Psi_t$ and $\nabla_{\mathbf{x}}\Psi_t$, the optimal control takes a analytic

form as in eq.(9). Since $\Psi_t$ and $\nabla_\mathbf{x}\Psi_t$ are explicit functions of $\mathbf{x}_t$, the resulting control law is essentially different from the feedforward control in sampling-based path integral control frameworks [5, 6, 7, 10, 18] as well as the parameterized state feedback PI control policies [14, 8]. Notice that at current time step $t$, we update the control sequence $\hat{\mathbf{u}}_{t,...,T}$ using the presented forward-backward scheme. Only $\hat{\mathbf{u}}_t$ is applied to the system to move to the next step, while the controls $\hat{\mathbf{u}}_{t+\mathrm{d}t,...,T}$ are used for control update at future steps. The transition sample recorded at each time step is incorporated to update the GP model of the dynamics. A summary of the proposed algorithm is shown in **Algorithm** 1.

---

**Algorithm 1** Sample efficient path integral control under uncertain dynamics

---

 1: **Initialization:** Apply random controls $\hat{\mathbf{u}}_{0,...,T}$ to the physical system (1), record data.
 2: **repeat**
 3:     **for** t=0:T **do**
 4:         Incorporate transition sample to learn GP dynamics model.
 5:         **repeat**
 6:             Approximate inference for predictive distributions using $\mathbf{u}^{old}_{t,..,T} = \hat{\mathbf{u}}_{t,..,T}$, see (10).
 7:             Backward computation of optimal control updates $\delta\hat{\mathbf{u}}_{t,...,T}$, see (13)(14)(9).
 8:             Update optimal controls $\hat{\mathbf{u}}_{t,..,T} = \mathbf{u}^{old}_{t,..,T} + \delta\hat{\mathbf{u}}_{t,..,T}$.
 9:         **until** Convergence.
10:         Apply optimal control $\hat{\mathbf{u}}_t$ to the system. Move one step forward and record data.
11:     **end for**
12: **until** Task learned.

---

### 3.2 Generalization to unlearned tasks without sampling

In this section we describe how to generalize the learned controllers for new (unlearned) tasks without any interaction with the real system. The proposed approach is based on the compositionality theory [26] in linearly solvable optimal control (LSOC). We use superscripts to denote previously learned task indexes. Firstly we define a distance measure between the new target $\bar{\mathbf{x}}^d$ and old targets $\mathbf{x}^{dk}, k = 1, .., K$, i.e., a Gaussian kernel

$$\omega^k = \exp\left(-\frac{1}{2}(\bar{\mathbf{x}}^d - \mathbf{x}^{dk})^\mathrm{T}\mathbf{P}(\bar{\mathbf{x}}^d - \mathbf{x}^{dk})\right), \tag{15}$$

where $\mathbf{P}$ is a diagonal matrix (kernel width). The composite terminal cost $\bar{q}(\mathbf{x}_T)$ for the new task becomes

$$\bar{q}(\mathbf{x}_T) = -\lambda \log\left(\frac{\sum_{k=1}^K \omega^k \exp(-\frac{1}{\lambda}q^k(\mathbf{x}_T))}{\sum_{k=1}^K \omega^k}\right), \tag{16}$$

where $q^k(\mathbf{x}_T)$ is the terminal cost for old tasks. For conciseness we define a normalized distance measure $\tilde{\omega}^k = \frac{\omega^k}{\sum_{k=1}^K \omega^k}$, which can be interpreted as a probability weight. Based on (16) we have the composite terminal desirability for the new task which is a linear combination of $\Psi_T^k$

$$\bar{\Psi}_T = \exp\left(-\frac{1}{\lambda}\bar{q}(\mathbf{x}_T)\right) = \sum_{k=1}^K \tilde{\omega}^k \Psi_T^k. \tag{17}$$

Since $\Psi_t^k$ is the solution to the linear Chapman-Kolmogorov PDE (7), the linear combination of desirability (17) holds everywhere from $t$ to $T$ as long as it holds on the boundary (terminal time step). Therefore we obtain the composite control

$$\bar{\mathbf{u}}_t = \sum_{k=1}^K \frac{\tilde{\omega}^k \Psi_t^k}{\sum_{k=1}^K \tilde{\omega}^k \Psi_t^k} \hat{\mathbf{u}}_t^k. \tag{18}$$

The composite control law in (18) is essentially different from an interpolating control law[26]. It enables sample-free controllers that constructed from learned controllers for different tasks. This scheme can not be adopted in policy search or trajectory optimization methods such as [10, 11, 14, 17, 19, 20, 21, 22]. Alternatively, generalization can be achieved by imposing task-dependent policies [27]. However, this approach might restrict the choice of optimal controls given the assumed structure of control policy.

# 4 Experiments and Analysis

We consider 3 simulated RL tasks: cart-pole (CP) swing up, double pendulum on a cart (DPC) swing up, and PUMA-560 robotic arm reaching. The CP and DPC systems consist of a cart and a single/double-link pendulum. The tasks are to swing-up the single/double-link pendulum from the initial position (point down). Both CP and DPC are under-actuated systems with only one control acting on the cart. PUMA-560 is a 3D robotic arm that has 12 state dimensions, 6 degrees of freedom with 6 actuators on the joints. The task is to steer the end-effector to the desired position and orientation.

In order to demonstrate the performance, we compare the proposed control framework with three related methods: iterative path integral control [18] with known dynamics model, PILCO [17] and PDDP [19]. Iterative path integral control is a sampling-based stochastic control method. It is based on importance sampling using controlled diffusion process rather than passive dynamics used in standard path integral control [5, 6, 7]. Iterative PI control is used as a baseline with a given dynamics model. PILCO is a model-based policy search method that features state-of-the-art data efficiency in terms of number of trials required to learn a task. PILCO requires an extra optimizer (such as BFGS) for policy improvement. PDDP is a Gaussian belief space trajectory optimization approach. It performs dynamic programming based on local approximation of the learned dynamics and value function. Both PILCO and PDDP are applied with unknown dynamics. In this work we do not compare our method with model-free PI-related approaches such as [10, 11, 12, 14] since these methods would certainly cost more samples than model-based methods such as PILCO and PDDP. The reason for choosing these two methods for comparison is that our method adopts a similar model learning scheme while other state-of-the-art methods, such as [20] is based on a different model.

In **experiment 1** we demonstrate the sample efficiency of our method using the CP and DPC tasks. For both tasks we choose $T = 1.2$ and $dt = 0.02$ (60 time steps per rollout). The iterative PI [18] with a given dynamics model uses $10^3/10^4$ (CP/DPC) sample rollouts per iteration and 500 iterations at each time step. We initialize PILCO and the proposed method by collecting 2/6 sample rollouts (corresponding to 120/360 transition samples) for CP/DPC tasks respectively. At each trial (on the true dynamics model), we use 1 sample rollout for PILCO and our method. PDDP uses 4/5 rollouts (corresponding to 240/300 transition samples) for initialization as well as at each trial for the CP/DPC tasks. Fig. 1 shows the results in terms of $\Psi_T$ and computational time. For both tasks our method shows higher desirability (lower terminal state cost) at each trial, which indicates higher sample efficiency for task learning. This is mainly because our method performs online re-optimization at each time step. In contrast, the other two methods do not use this scheme. However we assume partial information of the dynamics ($\mathbf{G}$ matrix) is given. PILCO and PDDP perform optimization on entirely unknown dynamics. In many robotic systems $\mathbf{G}$ corresponds to the inverse of the inertia matrix, which can be identified based on data as well. In terms of computational efficiency, our method outperforms PILCO since we compute the optimal control update analytically, while PILCO solves large scale nonlinear optimization problems to obtain policy parameters. Our method is more computational expensive than PDDP because PDDP seeks local optimal controls that rely on linear approximations, while our method is a global optimal control approach. Despite the relatively higher computational burden than PDDP, our method offers reasonable efficiency in terms of the time required to reach the baseline performance.

In **experiment 2** we demonstrate the generalizability of the learned controllers to new tasks using the composite control law (18) based on the PUMA-560 system. We use $T = 2$ and $dt = 0.02$ (100 time steps per rollout). First we learn 8 independent controllers using **Algorithm** 1. The target postures are shown in Fig. 2. For all tasks we initialize with 3 sample rollouts and 1 sample at each trial. Blue bars in Fig. 2b shows the desirabilities $\Psi_T$ after 3 trials. Next we use the composite law (18) to construct controllers without re-sampling using 7 other controllers learned using **Algorithm** 1. For instance the composite controller for task#1 is found as $\bar{\mathbf{u}}_t^1 = \sum_{k=2}^{8} \frac{\tilde{\omega}^k \Psi_t^k}{\sum_{k=2}^{8} \tilde{\omega}^k \Psi_t^k} \hat{\mathbf{u}}_t^k$. The performance comparison of the composite controllers with controllers learned from trials is shown in Fig. 2. It can be seen that the composite controllers give close performance as independently learned controllers. The compositionality theory [26] generally does not apply to policy search methods and trajectory optimizers such as PILCO, PDDP, and other recent methods [20, 21, 22]. Our method benefits from the compositionality of control laws that can be applied for multi-task control without re-sampling.

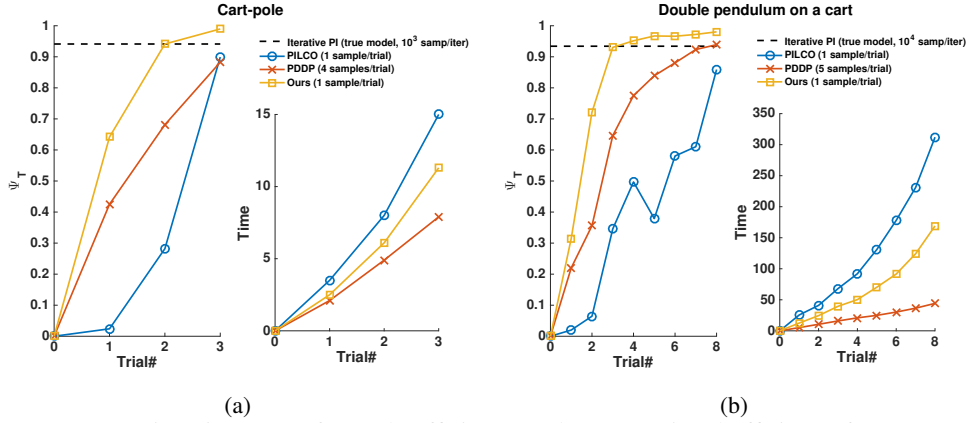

(a)                                              (b)

Figure 1: Comparison in terms of sample efficiency and computational efficiency for (a) cart-pole and (b) double pendulum on a cart swing-up tasks. Left subfigures show the terminal desirability $\Psi_T$ (for PILCO and PDDP, $\Psi_T$ is computed using terminal state costs) at each trial. Right subfigures show computational time (in minute) at each trial.

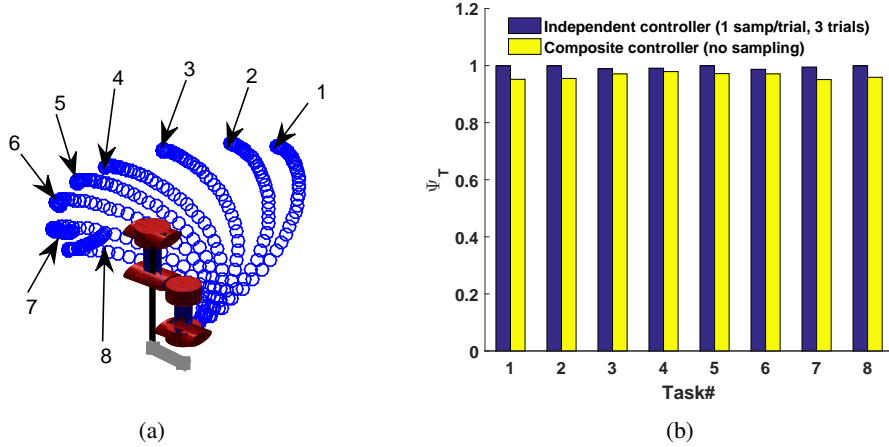

(a)                                              (b)

Figure 2: Resutls for the PUMA-560 tasks. (a) 8 tasks tested in this experiment. Each number indicates a corresponding target posture. (b) Comparison of the controllers learned independently from trials and the composite controllers without sampling. Each composite controller is obtained (18) from 7 other independent controllers learned from trials.

## 5   Conclusion and Discussion

We presented an iterative learning control framework that can find optimal controllers under uncertain dynamics using very small number of samples. This approach is closely related to the family of path integral (PI) control algorithms. Our method is based on a forward-backward optimization scheme, which differs significantly from current PI-related approaches. Moreover, it combines the attractive characteristics of probabilistic model-based reinforcement learning and linearly solvable optimal control theory. These characteristics include sample efficiency, optimality and generalizability. By iteratively updating the control laws based on probabilistic representation of the learned dynamics, our method demonstrated encouraging performance compared to the state-of-the-art model-based methods. In addition, our method showed promising potential in performing multi-task control based on the compositionality of learned controllers. Besides the assumed structural constraint between control cost weight and uncertainty of the passive dynamics, the major limitation is that we have not taken into account the uncertainty in the control matrix $\mathbf{G}$. Future work will focus on further generalization of this framework and applications to real systems.

### Acknowledgments

This research is supported by NSF NRI-1426945.

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
