[Reviews · NeurIPS 2015]

Submitted by Assigned_Reviewer_1

Summary

The paper presents a new method for path integral control. The proposed method leverages that the system control

matrix G is often known, and the uncontrolled dynamics (or dynamics under a reference controller) can be learned

using a Gaussian Process. The constraint on the reward function takes a more general shape than in previous PI

approaches which means among others that noise and controls can act in different subspaces. The authors also show

how their framework can be used for generalizing from known to new tasks, and evaluate the method on three simulated

robotics problems, where their method compares favourably to SOTA reinforcement learning and control methods.

Quality

Generally, the derivations seem correct and principled (although I'm unsure about the task generalization, see below).

Related work is sufficiently discussed, the authors point out the similarities and differences to related work and

compare to state of the art methods. The experiments are convincing, altough some details are missing (see below).

The task generalization seems odd: (13) states that the exponent of the reward functions will be linearly combined

to yield the exponent of the reward function of the new task. However, it's not clear to me how that results in the

exponent of the value function being the same kind of linear combination (14). In any case, this should be explained,

but I think that actually there might be an error here ((14) also states psi is the exponent of the cost function,

which holds only for the last time step as shown in the equation around line 247).

As defined by the authors, the task

generalization is specific to the case that the task is defined by a target, although it seems that other

task variations would be possible as long as a suitable task similarity kernel can be defined.

The experiments considers relevant and realistic tasks, and compare to SOTA methods. The technical details are a bit sparse

in the experimental section: the paper should mention what the dynamics and reward functions are, and how long sampled trajectories

are, in order for the experiment to be reproducible. (Possibly in the appendix). One baseline is iterative PI control,

is this one of the methods in table 1? It would be insightful to see how it compares on these qualitative aspects.

The comparison plots in Figures 1 and 2 should show error bars. The reported score is the exponent of the value fc, psi.

Is this the psi as calculated by the robot? What if the robot is overly optimistic in its calculation of psi? It would

be more objective to report the average (or cumulative (discounted)) reward (that is also what the algorithm sets out to optimize).

Clarity

Generally, the paper is well-written and explains the proposed methods rather well. The comparison to other methods, e.g.

in Table 1, helps understanding the relationship to the SOTA. There are a couple of grammatical and formatting errors,

see below under Minor points. One confusing point is that the GP has different regression targets in (2) (line 094) and line 208,

which doesn't include the reference controls. This should be made consistent or explained. The authors should explain how

the hyperparameters of the GP are set. It's unclear what's meant by "the posterior distribution can be obtained by

constraining the joint distribution" -- is conditioning meant?

Algorithm 1 is confusing - here, it looks as if an open-loop

control sequence is optimized while the rest of the paper discusses learning a feed-back controller.

If it's really just an open-loop sequence u_1 ... u_T that's returned in line 13 of the algorith, how can the algorithm deal

with the noise? This should

be clarified or corrected.

Originality and Significance

Although the main ingredients of the proposed algorithm have been used in earlier algorithms (propagation through an learned

stochastic forward model as in [15], path integral control), the algorithm combines these in a novel way. As far as I know,

this is an original formulation that seems to attain very good performance. By evaluating on realistic problems against strong

benchmarks, the proposed algorithm can be concluded to be a significant improvement over prior work.

Comments on the rebuttal:

Thanks for clarifying (14). Still, it seems there is a typo and Psi should be Phi (or Psi_t+dt should be included in the middle part). To me, it's therefore unfortunately still unclear what's meant here. I changed my confidence

& score accordingly. Thanks for clarifying the open-loop part, I would really change the notation here to avoid confusion.

I'm also still confused about what meant by the psi in figure (1), as I understand it this is the exponent of the final costs, but average cost would be more standard. Maybe the average cost could additionally be reported in the supplementary material?

Minor points

* Although generally well-written, the paper has some grammar issues. I recommend letting someone proofread it

(line 040 "has been existed", line 143 (while -> in contrast ?), line 231 - comma instead of full stop or reformulate,

line 345 is based on ... model -> are based on ... models

)

* the authors should avoid creating empty sections (e.g. between sec 2 and sec 2.1)

* there are some formatting issues (margins in line 128, table 1, brackets not scaled correctly in (8))

* would be clearer if the authors state how to obtain \tilde{G} from G

* line 192 table.1. -> Table 1.

* notation: in (13) x_t is written boldface on the right-hand side but italic on the left-hand side.

* line 339: Pilco requires an optimizer for policy evaluation -> is policy improvement meant?

* line 332 "6 actuators on each joint" -> is "one actuator per joint" meant?
Summary: The paper presents a new method for path integral control. The method is evaluated in convincing experiments. Possibly, there is an issue with the derivation for the multi-task setting, I would like to see the author's reply on this point.

Submitted by Assigned_Reviewer_2

The authors propose a novel formulation of path integral control, where they use an exponential form of the value function, which allows them to solve the path integrals of the cost analytically, and in particular they can evaluate the path integrals backwards. This gives them the ability to optimize using a forward-backward approach where they use a GP to model the paths forward in time, and then run the backward integration to get the path costs and derivatives, and then use a numerical technique to update the controls.

Quality: The quality of the paper is relatively high, in technical correctness. I enjoyed reading the paper and learned something new.

I was surprised that this technique outperformed iterative PI - unless I am missing something, this is almost certainly a problem with the number of samples (10^3 for cart-pole, 10^4 for double pendulum) which seem relatively low. No doubt the new technique is more efficient, but the authors need to be careful with their baseline evaluation.

Clarity: The major comment I have about the paper is that the authors move quickly through some key steps. In particular, the move from eqaution 7 to 8 is non-trivial, and is key to the overall paper. Similarly, the moves from the initial integral of \Psi_t (mid-page 5) to the final recursive form of \Phi_{t+dt)\Psi_{t+dt} is even harder.

Originality: The use of a GP for the model is not new, but the forward-backward is, additionally the ability to use the GP to infer the integral for a changed value function is at least new to me.

Significance: An interesting and substantial improvement to existing path integral methods.

Please number all equations -- I understand there is a school of thought that equations should only be numbered if they are referenced in the text, but that stylistic conceit creates major problems when others want to refer to the equations outside the text of the paper, either in reviews (like this one), in other papers referencing this one, or in discussion.
Summary: The authors have proposed a novel method that advances the state of the art. Overall a good paper, although not easy to read in places. This is largely due to page limitations so I can't hold the authors too much to blame for this.

Submitted by Assigned_Reviewer_3

The paper presents a novel PI-related method for stochastic optimal control. The main idea is to use a Gaussian process to model the trajectories resulting of applying a proposal control $u^old$ on the uncontrolled dynamics. The proposed Bellman equation is defined to optimize for $\delta u$, the control update at each iteration. The method takes advantage of analytical methods for GP inference to update the GP parameters efficiently and it is evaluated experimentally in three simulated tasks.

The idea is interesting and authors seem to have a working method that can compete with existing ones. I like the paper in general, but I think the following points need to be addressed:

The problem formulation section is a bit confusing, since there are two optimizations involved: one is the optimization over $u$ to find the optimal control and the other is the one that optimizes $\delta u$. From 2.1, it seems that the optimal u is given together with a proposal control u^old, which is clearly not the case, since finding the optimal u is exactly the main task. Authors need to clarify this point.

The organization of the paper could be significantly improved: since the proposed approach is not introduced in 3.1 but earlier, I would move subsection 3.1 after subsection 2.1. This would also improve the readability, since Eq (9) involves the inference step explained in subsection 3.1.

I also found confusing section 3.1, in which the GP is defined (line 208) for the uncontrolled process (u=0) which contradicts the definition in Eq(2) which depends on the current proposal control. Additionally, if the GP is initialized using sampled data from the uncontrolled process, authors should also clarify how the poor conditioning of the uncontrolled process in terms of effective sample size affects this initialization.

Also related to how the GP model is updated, it looks like the formulation ignores the cost of the current control sequence, see Eq(5). In iterative PI control this term enters in the equation as a Radon-Nikodim derivative [16,17]. Can the authors point at the equivalent term in their formulation? Can we still talk about a generalization of PI control or is this another policy search, PI-related method?

The use of belief propagation for forward inference may fail when the dynamics around the reference change abruptly, i.e. they are non-Gaussian. This happens in the presence of obstacles, for example, which does not seem to occur in the presented experiments. Authors should clarify if this is a potential problem or not.

About notation, although authors present the method as model-based, it is not clear that a model of the dynamics is used (they only have a probability distribution over trajectories via the GP). I suggest different notation for clarity.

Authors do a good job relating their approach with existing PI-related methods, a difficult task given the (already large) current literature on PI control. I have the following remarks in that respect:

- In PI^2 [8] the constraint is not ignored, but PI^2 is an open-loop policy search method, so the constraint is meaningless and the same argumentation of PI^2-CMA holds. - Feedback PI [17] uses a parameterized feedback term (the second order correction), but not a parameterized policy. - PI-REPS [11] is model-based (although the model can be learned, as in guided policy search [23]) - Iterative PI [16] is original PI with importance sampling and therefore: model-based, with same structural constraint as PI and without parameterized policy.

These remarks do not involve table 1 only, but text spread through the whole paper.

minor:

line 120: minimize(s) lines 144-145: sentence unclear: "while (...)." line 146: (to) act line 229: based on (a) probabilistic
Summary: The paper presents a novel PI-related method for stochastic optimal control that uses a Gaussian process to model trajectories. The approach is promising but it is not clear if this is a generalization of PI control or it is a policy search, PI-related method. Presentation should be improved and some other points need to be clarified.

Author Feedback
Author rebuttal: We thank all reviewers for their constructive comments. We will improve the manuscript as suggested.

Reviewer_1
As the reviewer pointed out, it is straightforward to see eq.(14) holds for the final time step. Since (14) is the solution to a linearized HJB PDE (7), if the solution holds on the boundary (which is the final time step) then it holds everywhere. Moreover, the linear recursive form of \Psi (line 244-252) also leads to linear form in (14) from t to T. However for any nonlinear HJB PDE this is not the case. In our particular case this linear combination is valid. We will clarify this point.
The iterative PI used as the baseline is the one in [16] under known dynamics. The standard PI in table 1 provides a theoretical framework but is not directly applicable to highly under-actuated systems such as DPC in our experiment.
The optimized control sequence in algorithm 1 is not an open-loop sequence. Each controller is a function of the current state/future beliefs (eq.9). We will clarify this.

Reviewer_2
The iterative PI is a sampling-based approach that relies on random explorations under known dynamics. It requires sampling at each iteration.
From (7) to (8) we applied the Feynman-Kac formula. We will reorganize some of the derivations.
We will number all equations.

Reviewer_3
In our iterative scheme we solve a control problem at each iteration by optimizing \delta u (Bellman eq. 5). Mathematically our problem is defined differently from a standard optimal control problem. We will further clarify this.
In regard to the organization. Sec.2 introduces a theoretical framework and in Sec.3 we design an algorithm. But we will take into account the reviewer's opinions on the reorganization.
Thanks for pointing out line 208, we will fix the typo.
Regarding the control cost in our iterative scheme, the reviewer has a valid point, but it appears that there is a misunderstanding. Our iterative control problem is defined differently from the iterative PI based on importance sampling. At each iteration the control cost matrix is adapted based on uncertainty of the dynamics. While the iterative PI solves the same control problem with a fixed criterion by iteratively sampling from different distributions. Our iterative scheme is not a generalization of importance sampling. It is more general since we solve a HJB PDE under uncertain dynamics. We will clarify based on your suggestions.
Our method is based on the assumption of Gaussian systems (line 92-93). Therefore we did not use non-Gaussian dynamics in our experiments. We agree with the reviewer that for non-Gaussian systems there may be a potential problem. We will discuss this issue in the final version.
The proposed method is model-based since we model the transition probabilities of the dynamics using GP inferences. We will reconsider the notations.

Reviewer_4
The reviewer stated the main advantage of PI is obtaining optimal policy without approximations. We would like to remind that exact solution to PI control exists for systems for which the Backward Chapman-Kolmogorov PDE has an analytical and therefore exact solution. For general nonlinear systems the sampling-based policy is an approximation up to order of dt^2. However for uncertain dynamics the case is different. In our method, the path integral (8) and its gradient can be computed analytically thanks to the GP representations of the transition probabilities, this lead to explicit feedback control laws.
In regard to comparing our method with "algorithms using approximations such as iLQG-based approaches". The PDDP algorithm [18] used for comparison is one of the aforementioned algorithms. We mentioned in line 343-345 that we did not numerically compare our method with S.Levine's work because of different model learning schemes. Our method and the methods for comparison (PILCO, PDDP) use GPs to learn dynamics, therefore the performance differences are due to the controller designs rather than the learned models.
The reviewer suggested that we have to show the computation speed of our method is significantly faster than sampling based approaches. There seems to be a misunderstanding. The point of this method is "sample efficiency" under uncertain dynamics. We did not claim that our method is faster than other sampling-based approaches. There are definitely faster methods but need more samples. The two methods we used for comparison are the state-of-the-art in terms of sample efficiency. In addition we do not sample from GPs. It is known that sampling functions from GPs could be very challenging (significantly different from sampling random variables). We will work on further comparisons with methods such as PI-REPS in an extended version of this work.
We mentioned in line 321 that the composite control law is different from an interpolation. Detailed discussions can be found in Todorov's work [22].

Reviewer_5 and 6
We thank the reviewers for their comments.